# Gaps in Accessibility of Pediatric Formulations: A Cross-Sectional Observational Study of a Teaching Hospital in Northern Thailand

**DOI:** 10.3390/children9030301

**Published:** 2022-02-22

**Authors:** Prangthong Tiengkate, Marc Lallemant, Pimlak Charoenkwan, Chaisiri Angkurawaranon, Penkarn Kanjanarat, Puckwipa Suwannaprom, Phetlada Borriharn

**Affiliations:** 1The Public Health Promotion, Research and Training Foundation (PHPT), Chiang Mai 50000, Thailand; prangthongg@gmail.com; 2Faculty of Pharmacy, Chiang Mai University, Chiang Mai 50200, Thailand; puckwipa.suwan@cmu.ac.th; 3Institut de Researche pour le Developpement (IRD) Collaboration, Faculty of Associated Medical Sciences, Chiang Mai University, Chiang Mai 50200, Thailand; 4Department of Pediatrics, Faculty of Medicine, Chiang Mai University, Chiang Mai 50200, Thailand; pimlak.c@cmu.ac.th; 5Department of Family Medicine, Faculty of Medicine, Chiang Mai University, Chiang Mai 50200, Thailand; chaisiri.a@cmu.ac.th; 6Health and Medicine Policy Center (HMPC), Faculty of Pharmacy, Chiang Mai University, Chiang Mai 50200, Thailand; 7Faculty of Allied Health Sciences, Northern College, Tak 63000, Thailand; phetladabor@gmail.com

**Keywords:** pediatric formulation, availability, accessibility, acceptability, quality, national medicine list, children, off-label, extemporaneous

## Abstract

The lack of appropriate medicines for children has a significant impact on health care practices in various countries around the world, including Thailand. The unavailability of pediatric medicines in hospital formularies causes issues regarding off-label use and extemporaneous preparation, resulting in safety and quality risks relating to the use of medicines among children. This research aimed to identify missing pediatric formulations based on the experience of healthcare professionals in a teaching hospital in northern Thailand. A cross-sectional survey was conducted to collect data on missing pediatric formulations, the reasons for their inaccessibility, their off-label uses, their reactions to the situation, and suggestions to improve access to these identified medications. The survey was distributed to all physicians, nurses, and pharmacists involved in prescribing, preparing, dispensing, and administering pediatric medicines. A total of 218 subjects responded to the survey. Omeprazole, sildenafil, and spironolactone suspension were most often identified as missing formulations for children by physicians and pharmacists. They are unavailable on the Thai market or in any hospital formulary. For nurses, sodium bicarbonate, potassium chloride, and chloral hydrate were the most problematic formulations in terms of preparation, acceptability, and administration. These medicines were difficult to swallow because of their taste or texture.

## 1. Introduction

The lack of medicine formulations suitable for children is a global problem, particularly in developing countries with fewer healthcare resources [1]. The unavailability of appropriate pediatric formulations suitable for administration obligates health care providers to order the local preparation and administration of unlicensed formulations by the manipulation of adult medicines [2]. There are several issues with the use of unlicensed medicines or compounded preparations, such as medicine stability and prescribing practices with limited data regarding their bio-drug, efficacy, and toxicity [3]. Many prescribers are forced to prescribe medicines off-label for pediatric patients due to the lack of child-specific formulations and approved indications for this population [4]. A study in Korea, for example, found that prescribing off-label/unlicensed medicines was independently associated with adverse drug events in pediatric patients [5]. Over the years, the US Food and Drug Administration and the European Medicine Agency have developed guidelines emphasizing the need for specific medications for pediatric patients, the need to reduce off-label use, and the importance of the acceptability and appropriate use of pediatric drugs [6]. These efforts have resulted in a sharp increase in the number of pediatric drug studies in Europe and the US. However, there is limited research on the development of child-appropriate formulations for medicines already available on the market due to a lack of financial incentives [7]. The development of pediatric medicines is complex from a clinical pharmacology standpoint because drug absorption, distribution, metabolism, and elimination among children differ from those in adults. Pharmacodynamics may also differ due to differences in drug receptors, which explains the variation between children and adults in terms of the efficacy and/or safety of medicines [1,5].

The gap in access to medicines for pediatric patients is a problem in many countries globally, including Thailand. One study in Korea, for example, showed that more than 73.5 percent of pediatric patients admitted to hospitals received off-label medicines [5]. In a study of pediatric patients with chronic disease, more than one-third of patients rejected oral formulations because of palatability issues in terms of smell, taste, or texture [8]. Reformulating, diluting, flavoring, and crushing adult medicines to facilitate their use in children are inevitable. These processes may lead to dosage errors, where too-high or too-low exposures to active ingredients may affect their efficacy and safety, as well as treatment acceptance [9].

Although the issue of the lack of medicine formulations for pediatric patients has received much attention internationally, little supporting evidence has been presented from the perspective of healthcare providers. The present study aimed to explore gaps in access to pediatric medicines, focusing on the experience of pediatricians, nurses, and pharmacists working in a teaching hospital in Thailand.

## 2. Materials and Methods

We carried out a cross-sectional descriptive study using a questionnaire to capture healthcare practitioners’ opinions on access gaps for pediatric drug products. After providing brief information about the research objectives and the expected benefits, the questionnaire gathered information about the participants’ general characteristics, such as pediatric subspecialty, age, and the duration of the caregivers’ experience with children. Then, open-ended questions were used to explore the participant’s experience of gaps in access to pediatric medicines. Pediatricians and pharmacists were asked about needed but unavailable medicines in the hospital formulary, the off-label use of medicines, and medicines considered problematic in terms of the difficulty for prescribers to determine the right dosage, their acceptability to children, and the complexity of their preparation or administration by caregivers. Nurses were asked about the preparation, administration, acceptability, and storage of pediatric medicine. Lists of the questions asked are shown in Table 1. For each product mentioned, further explanations were asked for. All medical personnel, pediatricians, pharmacists, and nurses practicing at the site and involved in pediatric care, both in the inpatient and outpatient services, were invited to participate. The study site was the largest teaching hospital in northern Thailand, which serves as a tertiary referral hospital for the region’s public hospital network. Severe and complex cases beyond the capacities of the district and provincial hospitals are referred to pediatric specialists at this hospital.

Questions were developed based on a review of the literature and informal interviews with health care practitioners. They were checked for validity and reliability by three experts: A specialist in pediatric medicine, a pediatric nurse, and a pharmacist. They were also pretested before the start of the study.

Data were collected from February to March 2021. Questionnaires were distributed in paper and electronic formats. We used the STATA Program [10] to analyze the data using descriptive statistics in terms of percentage, mean, and standard deviations.

The research protocol was approved by the Human Research Ethics Committee of the Faculty of Medicine, Chiang Mai University. All healthcare professionals provided written informed consent.

## 3. Results

A total of 223 participants, consisting of 36 doctors, 18 pharmacists, and 169 nurses, responded to the questionnaire. The specialties of the participants are presented in Table 2. General practitioners and pediatricians in all subspecialties responded to the questionnaire, as well as 169 nurses, representing 63.2 percent of all respondents. Table 3 describes the working experience of the study participants. Nurses had been working for 12.9 years on average (SD = 10.2), while doctors and pharmacists had been working for averages of 9.1 (SD = 9.0) and 2.9 (SD = 3.8) years, respectively.

Physicians and pharmacists were asked to name the five medicines most needed for their pediatric patients but unavailable in the hospital medicine list. A total of 65 medications were mentioned by physicians (Table 4). The most frequently mentioned were omeprazole syrup, sildenafil suspension, and prednisolone syrup. When asked about their experience of prescribing off-label medicines, the participating physicians mentioned a total of 35 medications. The most frequently prescribed off-label medicines were clopidogrel for Kawasaki disease, giant aneurysms, and the prophylaxis of thrombosis in children, followed by celecoxib for children with severe pain from hemophilia.

When the participants were asked about the most frequently prescribed children’s medicines requiring extemporaneous preparation, 41 medications were mentioned, with the top two being omeprazole and sildenafil suspensions. Omeprazole is unlisted in the hospital formulary because this suspension is rarely used and very costly. Sildenafil suspension is unavailable in Thailand and is only available at the hospital as adult tablets. Therefore, medicine manipulation by crushing and mixing with liquid is generally prescribed. Extemporaneous preparations of adult dosage forms make it difficult for children to accept their treatments and for doctors to adjust the dose. The most problematic pediatric drugs mentioned were warfarin and enoxaparin, which require close follow-up and regular blood-level monitoring.

Children’s medicines considered missing by the pharmacists are shown in Table 5. A total of 28 formulations missing from the hospital formulary were mentioned by 18 pharmacists. The most frequently mentioned drugs were erythromycin, cloxacillin, and hyoscine solution. Erythromycin solution, although listed on the National List of Essential Medicines (NLEM), was recently removed from hospital formularies because the Department of Health’s regulation limits the total number of medicines in each hospital. It was noted that cloxacillin powder for use in suspensions is available on the market, but only the sterile powder used for injection is listed on the NLEM and, therefore, able to be purchased by public hospitals. Hyoscine syrup is listed on the NLEM but is not available in hospital formularies. Regarding off-label use, the medicines most frequently mentioned by pharmacists were acetazolamide and topical nitroglycerine. Acetazolamide is prescribed for metabolic alkalosis and topical nitroglycerine, although not approved for use in children, is prescribed to increase local blood flow.

The extemporaneous preparations most frequently mentioned by pharmacists were omeprazole, furosemide, and sildenafil suspensions (also reported as missing by physicians). These medicines are neither available in child-appropriate forms on the market nor included in the NLEM. The medicines considered most difficult to prescribe for children were amoxycillin suspension, paracetamol syrup, and aspirin (film-coated tablet) because of concerns regarding preparation, dosing, and product manipulation.

Nurses working in the pediatrics department were asked to name the most problematic children’s medicines in terms of preparation, administration, and storage, as well as acceptability by children or caregivers. A total of 166 nurses responded to the survey. As shown in Table 6, the medicines considered most problematic in terms of preparation were 7.5 percent sodium bicarbonate, injectable furosemide, omeprazole, and calcium polystyrene sulfonate (Kalimate^®^). Nurses reported that ampules of 7.5 percent sodium bicarbonate were difficult to break open safely. The same problem was mentioned for furosemide injections. Nurses reported having to mix the content of capsules of poorly water-soluble omeprazole with orange juice to increase their solubility before their administration to children through a nasogastric tube. Calcium polystyrene sulfonate (Kalimate^®^) was noted because it does not disperse well and presents a bad taste when prepared extemporaneously.

Asked about acceptability problems by children in terms of taste, texture, appearance, and swallowability, the most frequently mentioned medicines were potassium chloride, multivitamin drops, and chloral hydrate. Potassium chloride tastes bland and requires a large volume of liquid. The medicine leaves a burning sensation in the mouth and stomach. Multivitamin drops have a pungent odor and sticky texture. Chloral hydrate was reported to have a bitter taste and sticky texture. Babies reject or refuse these medicines, often resulting in inappropriate dosing.

With regard to storage, the expensive micafungin was reported as the most problematic medication, as the injectable is used in small quantities, and the remaining portion needs to be refrigerated. The medicine needs to be kept away from light and prepared in a sterile environment. Insulin was also mentioned, as it requires temperature control, presenting difficulties for patients without access to their own refrigerator.

## 4. Discussion

Managing children’s medicines remains a challenge for healthcare practitioners. Off-label and unlicensed medicine use due to a lack of age-appropriate formulations is a global concern [11,12,13]. A study conducted in Norway reported that 91 percent of pediatric patients received at least one off-label or unlicensed drug prescription [13]. Similarly, in a study conducted at a Spanish teaching hospital, 90.2 percent of patients in the neonatal intensive care unit received at least one off-label prescription [14]. In a pediatric department of tertiary care hospitals in Pakistan, the prevalence of off-label medication use was 52.1 percent, while the prevalence of the unlicensed use of medicines was 33.4 percent [15]. This practice is common, although physicians and pharmacists acknowledge these issues [16,17]. Efficacy and safety are their major concerns [16,17].

Results from this study show that physicians have difficulties in prescribing appropriate formulations for their pediatric patients for many reasons. The needed child-appropriate formulations may not be listed on the hospital formulary because of the low number of patients requiring treatment. Some child-appropriate medicines may also be available but not on the Thai market—e.g., the suspension of sildenafil mentioned by some respondents. Some children’s medicines are commercially available, but too expensive for the hospital budget constraints, and only medicines in the forms of adult tablets or preparations suitable for injection may be listed in the hospital formulary. Erythromycin, cloxacillin, and hyoscine are good examples of this. When medicines are only available in the adult tablet form, product manipulation, such as crushing and mixing with a liquid vehicle, are common practices [9,18,19,20]. A study on the manipulation of medicinal products for pediatric patients carried out at a German university hospital showed that 57 percent of patients received manipulated medicine products, and 71 percent of these preparations were provided because an appropriate medicinal product was commercially unavailable [21]. Extemporaneous preparations of adult dosage forms by pharmacists have to be also performed. This makes it difficult for nurses or caregivers to prepare and administer the medicines, for children to accept them, and for doctors to adjust dosage overtime to ensure appropriate safety and effectiveness [20,22].

Pediatric formulations may be unavailable at a hospital because they have not been included in the National List of Essential Medicines (NLEM). Depending on their size, public hospitals have to select 70 to 100 percent of the products listed in their formulary from the NLEM. Importantly, in Thailand, there is no NLEM for children. The hospital drug committee selects the medicines to be included in the formulary based on clinical needs and the number of cases to be treated, which includes cases in both adults and children. As a result, many children’s medicines are never selected or have even been eliminated from hospital formularies. Even when some medicines are on the NLEM, they may not be included in a hospital formulary if they are costly and the hospital budget is limited [23]. Therefore, hospital pharmacists must make extemporaneous preparations for pediatric use—e.g., omeprazole or sildenafil suspensions. Figure 1 illustrates gaps in the accessibility of pediatric formulations in Thailand. Although the hospital pharmacy has a specific department for extemporaneous preparations, there are no up-to-date standards for preparing children’s medicines, leading to fears of instability and forcing patients to return frequently to the hospital to refill their medicines.

Where formulations are missing, many important medicines have to be used off-label with limited supporting evidence. Pediatricians, particularly specialists, have reported the frequent use of off-label medicines for their pediatric patients in terms of indication, dosage, or administration route [11,12,14,17,24,25].

Another problem is related to medicine acceptability. Taste, smell, appearance, texture, and ease of swallowing affect drug administration and intake. Children’s refusal to accept drugs affects the amount they receive and leads to poor treatment outcomes [26,27]. Usability is another issue. In this study, we found that ampules of injectable products were sometimes difficult to break, and preparation was dangerous for health care professionals or caregivers.

Storage is a particular problem for medications that are temperature- and light-sensitive [28]. Temperature-sensitive medicines are difficult for patients to transport and store at home. Targeted education sessions are necessary for patients and caregivers to ensure that temperature changes do not affect the efficacy and stability of drugs.

The results of this study should alert practitioners and policymakers to the gaps in access to children’s medicines at the hospital level, which are a consequence of the lack of national policies on children’s medicines. This study has several limitations. The generalizability of our findings is limited because our study was conducted in a single teaching hospital in northern Thailand. Lists of medicines mentioned by practitioners may vary by region and the level of specialization/treatment capacity of hospitals. The doctors interviewed had a variety of specialties and sub-specialties, resulting in them noting medicines of interest to them and identifying the problematic medicines most relevant to their own experience. Future research should apply this study’s findings to develop a structured survey to assess gaps in accessing children’s medicines in different hospital settings. In this study, we learned that perceptions of the medicine gap differed according to the respondent’s profession. The physicians were concerned about medicines needed for treatment, the pharmacists focused on the availability of medicines and extemporaneous preparation, and the nurses worried about the preparation and acceptability of the products. Previous studies have shown diverse levels of concerns among health professionals, with higher concerns found among pharmacists [16,17]. Therefore, exploring gaps in pediatric medicine from multiple perspectives would allow us to learn more about the situation as a whole and get a broader picture. More evidence is urgently needed to raise public awareness on access gaps and the quality use of medicines for children.

## 5. Conclusions

Gaps in the effective use of medicines in children need to be addressed. Availability, accessibility, quality, and acceptability are key aspects to consider. Many important medicines are prepared as extemporaneous products with a short shelf-life and questionable levels of acceptability. Some medicines are difficult to prepare and administer, leading to problems with appropriate dosage. The development of a National Medicine List for Children would be an important step towards ensuring that pediatric patients have access to appropriate medicines. This would facilitate the inclusion of pediatric medicines in the hospital medicine list and would encourage the national pharmaceutical industry to produce these medicines commercially, thus ensuring that high-quality pediatric medicines with good levels of acceptability are available on the market.

## Figures and Tables

**Figure 1 children-09-00301-f001:**
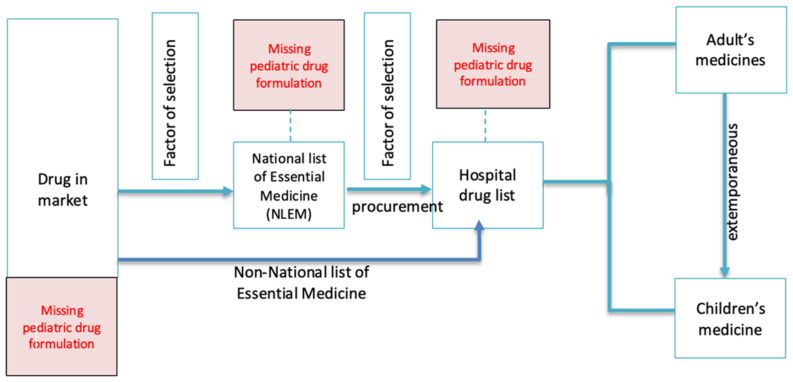
Gaps in the accessibility to pediatric formulations in Thailand.

**Table 1 children-09-00301-t001:** Questions asked about access gaps of pediatric medicines.

Questionnaire for Physicians and Pharmacists	Questionnaire for Nurses
1. Which needed medicines are unavailable in your hospital?	1. Which are the most problematic medicines to prepare?
2. Which off-label medicines do you frequently use among children?	2. Which are the most problematic medicines in terms of acceptability?
3. Which medicines require extemporaneous preparation?	3. Which are the most problematic medicines to administer?
4. Which are the most problematic medicines?	4. Which are the most problematic medicines in terms of storage?

**Table 2 children-09-00301-t002:** Specialization of study participants.

Specialization	Overall, *n* (%) (*n* = 223)
Physicians	36 (16.1)
Pediatrician (general)	12
Pediatric hematology-oncology	6
Pediatric cardiology	7
Neonatal and perinatal medicine	3
Developmental and behavioral pediatrics	3
Pediatric neurology	1
Pediatric gastroenterology and hepatology	1
Pediatric allergy and immunology	1
Pediatric Nephrology	1
Pediatric Rheumatology	1
Pharmacists	18 (8.1)
Inpatient services	8
Outpatient services	4
Extemporaneous preparations	4
Drug information services	2
Nurses	169 (75.8)
Pediatric Nursing Section	141
Neonatal Intensive Unit: NICU	20
Nurse administration	5
Milk preparation department	2
Child development	1

**Table 3 children-09-00301-t003:** Working experience of study participants.

	Working Experience, *n* (%) (*n* = 223)			
	<1 Year	1–5 Years	5–10 Years	>10 Years
Physicians	1 (0.4)	13 (5.8)	11 (4.9)	11 (4.9)
Pharmacists	4 (1.8)	11 (4.9)	1 (0.4)	2 (0.9)
Nurses	7 (3.1)	48 (21.5)	28 (12.6)	86 (38.5)

**Table 4 children-09-00301-t004:** Most needed children’s formulations are unavailable in the hospital, as mentioned by physicians.

Children’s Formulation	Number (%) of Responses
Missing formulations	
Omeprazole suspension	9 (7.1)
Sildenafil suspension	9 (7.1)
Prednisolone syrup	8 (6.3)
Propranolol injection	6 (4.7)
Eltrombopag syrup	5 (3.9)
Spironolactone suspension	5 (3.9)
Total	127 (100.0)
Off-label use	
Clopidogrel	5 (9.1)
Celecoxib	4 (7.3)
Alectinib	3 (5.5)
Aprepitant	3 (5.5)
Risperidone	3 (5.5)
Total	55 (100.0)
Extemporaneous preparations	
Omeprazole suspension	15 (10.9)
Sildenafil suspension	13 (9.4)
Aspirin	9 (6.5)
Ursodeoxycholic acid	9 (6.5)
Calcium carbonate	7 (5.1)
Prednisolone	7 (5.1)
Total	138 (100.0)
Problematic children’s medicines	
Warfarin	13 (19.7)
Enoxaparin	7 (10.6)
Heparin	4 (6.1)
Cyclosporin A	4 (6.1)
Aspirin	3 (4.5)
Total	66 (100.0)

**Table 5 children-09-00301-t005:** Most needed children’s formulations are unavailable in the hospital, as mentioned by pharmacists.

Children’s Formulation	Number (%) of Responses
Missing preparation	
Erythromycin solution	8 (14.8)
Cloxacillin solution	6 (11.1)
Hyoscine solution	6 (11.1)
Salbutamol syrup	4 (7.4)
Total	54 (100.0)
Off-label use	
Acetazolamide	7 (26.9)
Topical nitroglycerine	7 (26.9)
Cloxacillin	4 (15.4)
Sulfamethoxazole and trimethoprim	3 (11.5)
Total	26 (100.0)
Extemporaneous preparation required	
Omeprazole suspension	13 (18.3)
Furosemide suspension	12 (16.9)
Sildenafil suspension	8 (11.3)
Spironolactone suspension	8 (11.3)
Vitamin D suspension	8 (11.3)
Total	71 (100.0)
Problematic children’s medicines	
Amoxicillin suspension	9 (23.7)
Paracetamol syrup	9 (23.7)
Aspirin (film-coated tablet)	4 (10.5)
Azithromycin syrup	2 (5.3)
Total	38 (100.0)

**Table 6 children-09-00301-t006:** Most needed children’s formulations unavailable in the hospital, as mentioned by nurses.

Children’s Formulation	Number (%) of Responses
In medicine preparation	
7.5% Sodium bicarbonate	68 (13.1)
Furosemide injection	30 (5.8)
Omeprazole capsule	30 (5.8)
Calcium polystyrene sulfonate	27 (5.2)
Calcium tablet	25 (4.8)
Total	519 (100.0)
In children’s acceptability to medicines	
Potassium chloride	94 (21.1)
Multivitamin drops	40 (9.1)
Chloral hydrate	33 (7.5)
Sodium phosphate	32 (7.3)
Ceftazidime	21 (4.8)
Total	441 (100.0)
In administration	
Potassium chloride	79 (20.1)
Chloral hydrate	37 (9.4)
Sodium phosphate	24 (6.1)
Multivitamin drops	21 (5.3)
Dipotassium phosphate	20 (5.1)
Total	393 (100.0)
Problematic children’s medicines	
Micafungin injection	20 (11.3)
Insulin	13 (7.3)
Amoxicillin/clavulanic acid injection	12 (6.8)
Ampicillin	11 (6.2)
Ganciclovir	11 (6.2)
Total	177 (100.0)

## Data Availability

Not applicable.

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
