# Peer review of "Gaps in Accessibility of Pediatric Formulations: A Cross-Sectional Observational Study of a Teaching Hospital in Northern Thailand"

_children, 2022, doi:10.3390/children9030301_

Round 1
Reviewer 1 Report
1. Title: The study is a cross-sectional observational study and not a case study. The title should be modified to be compatible with the study design.
2. Linguistic polishing is required all over the text
3. Trade name of the pharmaceutical product (e.g., Bactrim) should be replaced by the scientific name in the table.
Reviewer 2 Report
In this manuscript, the authors approached an important topic regarding the problems that occur with the use of medicines in the treatment of children. The non-availability of appropriate pediatric formulations causes the use of some medicines with off-label indications or many extemporaneous preparations. This is a common practice worldwide, especially in developing countries.
The objective of the study was to evaluate current practice and to identify the missing pediatric formulations towards pediatric drugs use in Thailand (one teaching hospital -case study). Based on a questionnaire completed by representative healthcare professionals (physicians, pharmacists, nurses) the authors discuss the availability, accessibility and acceptability of pediatric medicines.
This study was limited to inpatient resource-utilisation data in one clinical center (teaching hospital -case study), but useful for the data – which could be generalized - focus on the conditions and needs in developing countries.
The suggestions for the authors are:
- Manuscript has only a few references regarding the medical practice in other countries. Include and compare the results with data concerning the experience from others countries and the conclusion from published reviews (topic: missing pediatric drugs / prescription of off-label medicines)
- Discuss the differences of opinion in identifying the missing formulations between the two groups - pharmacists and doctors (from the same hospital)
- Revise author instructions for correct references citation in the manuscript !!! (all References)
The English level is good.
I consider that this article could be accepted with minor revision
